# A Novel Approach for 3D Printing Fiber-Reinforced Mortars

**DOI:** 10.3390/ma16134609

**Published:** 2023-06-26

**Authors:** Dragoș Ungureanu, Cătălin Onuțu, Dorina Nicolina Isopescu, Nicolae Țăranu, Ștefan Vladimir Zghibarcea, Ionuț Alexandru Spiridon, Răzvan Andrei Polcovnicu

**Affiliations:** 1Faculty of Civil Engineering and Building Services, “Gheorghe Asachi” Technical University of Iaşi, 43 Mangeron Blvd., 700050 Iaşi, Romania; catalin.onutu@academic.tuiasi.ro (C.O.); dorina-nicolina.isopescu@academic.tuiasi.ro (D.N.I.); nicolae.taranu@academic.tuiasi.ro (N.Ț.); stefan.zghibarcea@holcim.com (Ș.V.Z.); ionut-alexandru.spiridon@student.tuiasi.ro (I.A.S.); razvan-andrei.polcovnicu@student.tuiasi.ro (R.A.P.); 2The Academy of Romanian Scientists, 3 Ilfov Street, Sector 5, 050663 Bucuresti, Romania

**Keywords:** 3D printing, fiber-reinforced mortar, fresh properties, hardened properties, additive manufacturing

## Abstract

Three-dimensional printing with cement-based materials is a promising manufacturing technique for civil engineering applications that already allows for the design and the construction of complex and highly customized structures using a layer-by-layer deposition approach. The extrusion mechanism is one of the most expensive parts of the 3D printer. Also, for low-scale 3D printers, based on the shape of the extruder and the geometry limitation of the mixing blade, the 3D mixture is often limited to a narrow range of materials due to the risk of layer splitting or blockage. Therefore, there is a need to develop affordable and feasible alternatives to the current design–fabrication–application approach of 3D printers. In this paper, various newly designed mixtures of fiber-reinforced mortars that can be 3D printed using only a commercially available screw pump are analyzed based on their fresh properties and mechanical characteristics. The results, in terms of extrudability, buildability, flowability, and flexural and compressive strengths, highlight the potential of using this technology for constructing complex structures with high strength and durability. Also, the reduced facility requirements of this approach enable 3D printing to be made more available for civil engineering applications. With further innovations to come in the future, this method and these mixtures can be extended for the sustainable and economically feasible printing of single-family housing units.

## 1. Introduction

Three-dimensional (3D) printing is a process that enables the manufacture of solid objects based on a digital model. It involves laying down successive layers of a material (e.g., plastic, mortar, clay, metal, or ceramic), starting from the base and continuing to the top part of the digital model, until the real object is completed [1]. Nowadays, 3D printing is used in a large variety of industries including civil and mechanical engineering, aerospace, and healthcare, transforming the way products are designed and manufactured [2,3,4]. The most notable benefits that this technology may offer are faster prototyping, faster manufacturing, reduced waste, increased quality, and high customization [5]. 

In the civil engineering domain, the most common material suitable for 3D printing is mortar [6,7]. The process usually consists of a mixture of cement, water and aggregate that is extruded through a custom print head and deposited in a precise manner to form the desired shape [8]. The 3D-printed structures can have exceptional strength (compression and tensile) and durability, making them suitable for various uses in construction and infrastructure applications [9,10]. When compared to the traditional construction methods, this technique offers several benefits including the ones mentioned above and the possibility to create unique and intricate designs, as well as the ability to print integrated components like electrical and plumbing systems directly into the structure [11,12].

In the construction industry, the implementation of additive manufacturing is intertwined with the pursuit of opportunities to enhance efficiency and profitability in this field [13]. Recently, 3D printing has been applied not only in constructing residential houses, but also in fabricating various structures and urban components. The latter includes, among others, remarkable constructions such as: a 3D-printed bridge in Shanghai made by Professor Xu Weiguo from the Tsinghua University (School of Architecture); the Dubai Municipality building; full-size, prefabricated houses created by Shanghai-based Winsun; the headquarters of the Dubai Future Foundation (DFF); the bridge over the Oudezijds Achterburgwal in Amsterdam; and 3D-printed homes in Austin, Texas, designed by Icon [14]. 

The 3D mortar needs to meet certain fresh-state properties of printability, pumpability, buildability, and open time, as well as have satisfactory mechanical characteristics when hardened [15,16]. In order to achieve these fresh-state properties, various admixtures like plasticizers, thickeners, and thixotropic agents have been used by researchers to balance the viscosity, shape retention, rheology, fluidity, thixotropy, and slumping of the 3D material [17,18,19,20]. Besides these admixtures, accelerators are frequently used to facilitate the fast solidification and early strength development of 3D mortars [21]. The accelerators used in the concrete industry and, in particular, in 3D printing, are either alkaline or alkali-free products [22]. Regardless of the type of the accelerators, the hardening rate and the early setting of the 3D mortars/concrete is increased in nearly all cases. However, due to the complex composition of 3D mortar/concrete, in some particular cases, certain accelerators may have compatibility issues with cement and other admixtures (e.g., water-reducing products) [23]. In these unique cases, the accelerators do not have any effect regarding the hardening of the material and may cause unstable setting times and serious late strength loss. Detailed research on the compatibility between concrete and various accelerators is presented by Su et al. in [24].

More recently, various researchers have been interested in the possibility of 3D printing alkali-activated materials, also known as geopolymers [25,26]. This is due to their ability to substantially diminish the high CO_2_ footprint inherent to classical concrete applications. Pointedly, geopolymers are two-part mixtures obtained by combining an alkaline solution with precursors made of aluminosilicate-rich and/or alumina materials [27,28]. Recently, a material called laterite was developed and used as a precursor with remarkable performance [29]. Extensive research about geopolymers and the feasibility of using them as a substitute for cement is presented by Su et al. in [30].

On the other hand, in order to ensure the ductility of the 3D-printed structures, as well as to fulfill certain conditions referring to the mechanical characteristics, various reinforcement methods have been proposed and analyzed by researchers [31]. Based on the literature review, some of the most investigated reinforcements that are suitable for 3D-printed mortars include polymer fibers (polypropylene, glass, and carbon), metallic fibers, steel bars (placed before or during the printing process), polymer or steel meshes (placed before or during the printing process), nails or short rebars (placed perpendicular to at least two consecutive layers of the printed material), U-shaped nails (placed in a specific pattern on each layer of material), tensioned strands (similar to reinforced concrete post-tensioned members), and steel cables (capable of following the printing path) [32,33,34,35,36,37,38,39,40,41,42,43,44,45,46,47,48,49,50,51].

All of the above-mentioned methods can be perfectly fitted to certain applications and printing devices, but it is difficult to apply them to other cases because of hardware limitations or particular structural requirements. Hence, for now, an optimum solution could be obtained by combining various alternatives and by developing either custom-made 3D mortar for certain printer configurations or a new hardware configuration for the existing 3D mortar mixtures. 

In this paper, various newly designed mixtures of fiber-reinforced mortars that can be 3D printed using commercially available screw pumps are analyzed based on their fresh properties and mechanical characteristics. The study involved the preparation of 30 mixtures using three distinct preparation methods, and 28 other mixtures after characterizing the preparation methods. The specimens printed with the larger nozzle size showed a greater susceptibility to eccentricities, resulting in significantly lower compressive and flexural strength values. Additionally, the compressive failure mechanism of the mixes printed with a narrow nozzle resembled the characteristic failure mode of traditional concrete specimens. In terms of mechanical properties, the values obtained for the specimens printed with the narrow nozzle (20 mm) are 32% higher for flexural strength (at 28 days) and 60% higher for compressive strength (at 28 days) compared to the ones determined for the specimens printed with the 45 mm nozzle.

The main objectives of this study were to design affordable 3D mortars (made with common materials) and to print these mixtures using reduced facility, thus allowing the 3D printing technology to be made more available for civil engineering applications. Also, three distinct mixing methods were developed, each of them providing identical results. In this manner, the necessary prerequisites for preparing the printing material were confirmed, both in a factory or laboratory setting and on-site at construction sites. These methods have been designed to cater to diverse requirements and operational conditions, enabling efficient and reliable material preparation. With further innovation in the future, this approach and these mixtures might be extended for the sustainable and economically feasible printing of single-family housing units.

## 2. Materials and Mix Design

In order to develop a 3D mortar mix with acceptable pumpability, printability, and workability firstly, the materials’ compatibility with the printer components should be checked, as should the storing, delivering, and printing conditions. A custom-made, small-scale gantry printer was utilized for the studies presented in this paper. The printer configuration is depicted in Figure 1. Although the small-scale printer was capable of producing high-quality results, future studies will require the use of a larger printer to accommodate bigger print jobs.

Natural quartz sand 0–1 mm; CEM II/A-S 52.5 R Portland slag cement, in line with CP 012-1:2007 [52], NE 013:2002 [53], GP 075:2002 [54], and ATE 004-07/1707-2022 [55]; and limestone filler from a local quarry basin were used as the main components for the preparation of the cementitious mortars. The CEM II/A-S 52.5 R Portland slag cement consists of 85% cement clinker, 12% blast-furnace slag, and 3% minor auxiliary components. The particle size distribution of the component materials is illustrated in Figure 2.

All mixtures were reinforced with 12 mm long monofilament polypropylene fibers with an equivalent diameter of 21–34 microns. The tensile strength of the fibers, according to the manufacturer, was ≥300 N/mm^2^ [56]. Two types of additives were used in every mixture: a viscosity-modifying agent consisting of an aqueous solution and a high-molecular-weight synthetic copolymer, and a superplasticizer that enables fast strength development at early ages of hydration at low ambient and heat-curing temperatures [57].

The mixtures studied in this paper can be prepared using 3 distinct methods, as indicated in Table 1. The mixtures’ components are indicated in Table 2. The experimental results presented in the following sections were obtained for mixtures prepared using method 1 (using a pan mixer with a constant speed). It should be noted that the mixtures were designed so the 3D mortar could be easily poured first through the pump feeder shaft, then to the outlet unit with the inside screw pump, and finally further through to the hose and nozzle (Figure 3). Thus, the maximum size of the selected ingredients was limited to one-tenth (1 mm) of the nozzle’s smallest dimension. 

As observed in Table 2, the properties of the fresh material are highly dependent on the quantity of additives introduced in the recipe. In Figure 4, the M28 mix appearance and texture before and after the addition of the additive and final mixing are illustrated.

## 3. Extrudability

Based on the material system and the feeder configuration, there are four types of 3D printers (Figure 5) [58,59,60,61,62,63,64]. In order to supply the mixed mortar to the nozzle, two types of feeders may be utilized: remote and local (Figure 5a,b,d). The remote feeder stores the 3D material and sends it to the printing nozzle or the local material bin via a transmitting pipe (Figure 5a,b). On the other hand, the raw extruding systems do not require a material feeder since the extruding mechanism serves as the material bin, as illustrated in Figure 5c.

For the printing of the fiber-reinforced mortars described in this work, a system composed of a material remote feeder, screw pump, hose, and nozzle was utilized (Figure 4a). Even though no extruding mechanism was used, in Table 2 the first analyzing parameter is termed “extrudability”. The latter is described as the process for the deposition of continuous layers of the mixture through a nozzle, according to a specific path. The term refers to a printing property being used for the characterization of both configurations (with and without an extruding mechanism). 

The extrudability issues cannot be adjusted during the printing process. The changes made to a mixture can be quantified only by referring to a looped process that occurs between the before and after of the addition or substitution of components. As a general rule, the mixtures with less stiffness are preferable for a smooth extrusion process (Figure 6). On the other hand, a mixture that is too viscous causes clogging during printing, like in the case of mixes M1–M16. 

In addition to the mixture viscosity, extrudability is also influenced by the cross-section shape and size of the nozzle, as well as the frictional adhesion stress between the printing material and the rotor, stator, and nozzle. In this respect, the rotor and the stator were sprayed with a silicone oil product to enable the lubricated engagement of the gears. As for the nozzles, all of the configurations (18, 20, 25, and 45-mm) were designed and printed specifically for this work using a commercially available 3D printer loaded with a recycled polylactic acid (PLA) filament (Figure 7) [65,66]. It was found that only the M27 mix was blocked at the nozzle level (for the diameters of 18-, 20, and 25 mm); the rest of the mixtures exhibited clogging in the pump feeder shaft or in the outlet unit. Two end accessories were designed and manufactured for the nozzles: one that ensures a waved surface between the layers (to increase the layer bonding), (Figure 8a) and one that ensures smooth lateral surfaces for the printed elements (Figure 8b). While the utilization of the second end-cap accessory is a matter of choice regarding the finishing texture of the printed element, the utilization of the first one does not significantly improve the overall behavior or any aspects of the printed elements.

The extrudability is also influenced by the separation of the materials in the hose due to insufficient mixing before the pumping stage is initiated. The mixing methods presented in Table 1 were developed by testing the other 30 mixtures that had been prepared by varying the order in which components were added and the mixing speed and time. Since these mixtures were used only for fine-tuning the preparation methods, they are not presented and discussed in the following parts of this paper. 

For all of the mixtures, it was found that pausing the printing process brings about visible changes in the materials’ rheological properties and causes the insufficient bonding of the layers due to the lack of surface moisture.

## 4. Buildability

Buildability is considered to be the paramount parameter for a printable mixture. By definition, a printed material with satisfactory buildability should hold its shape without excessive deformations and should have acceptable settlement in the bottom layers [67]. In order to develop such a material, all the mixtures indicated in Table 2 were slowly pumped through the system so as to visually observe which one has sufficient stiffness and quickly gains green strength to maintain a plane substrate for the upcoming layers. Figure 9 illustrates the difference between a mix (M10) with unsatisfactory buildability (local collapse by layer splitting) and a mix with high buildability (M26). 

The buildability test is a common evaluation for 3D-printed materials which allows us to determine the maximum number of layers that can be poured on top of each other without generating the collapse of the structure. For the mixtures with low or no buildability, the printed project collapsed either due to the progressive loss of stability (the bottom layers developed excessive deformation), (Figure 10a) or suddenly due to plastic failure (probably, the ultimate stress in the lower third of the structure exceeded the material yield strength). On the other hand, for the mixes with high buildability, the test provided satisfactory results that were independent of the nozzle dimension and end-cap configuration (Figure 10b,c).

In order to increase the buildability of 3D-printed mortars, the mixture can be supplemented with cement or silica powders. However, this kind of solution was not selected for this work due to the fact that the mixture’s total cost would have significantly increased. On the other hand, the “fine tuning” of the mixture, although more time consuming, may provide a cost-effective solution.

## 5. Flowability

Flowability is defined as the ability of a material to both flow and fill an available space under its own weight [68,69]. In 3D printing applications, flowability is considered crucial for achieving a consistent manufacturing process, since it contributes to the reduction of print defects and the prevention of blockages. Flowability is influenced by various factors, including the printing process parameters and the rheological properties of the 3D mortar. The latter refers to the viscosity and yield stress, which are parameters that characterize the flow behavior. Therefore, it is essential to control the flowability of the 3D-printed mortar to optimize the manufacturing process and to achieve high-quality printed objects/structures. In this respect, the slump and the slump flow test are two widely used methods to measure the flowability of 3D-printed mortars.

The slump test is a simple and common method to determine the flowability of fresh 3D mortars. The test involves measuring the slump, which is the deformation of the concrete when it is placed in a cone-shaped mold (Figure 11). 

The values listed in Table 2 were determined by performing tests in accordance with the European Standard EN 12350-2:2019 [70]. The slump was determined as the difference in height between the original height of the cone and the height of the 3D mortar after deformation (Figure 12). According to previous studies, the mixtures were fine-tuned to obtain values ranging from 40 to 60 mm [71]. 

The slump test has some limitations, which are particularly related to 3D-printed mortars. The test does not account for the shear-thinning behavior of the mortar, which can result in inaccurate measurements [72]. Shear-thinning is a phenomenon in which the viscosity of a material decreases as the shear rate or deformation increases. This behavior is common for 3D-printed mortars due to the high shear rates experienced during the printing process. Therefore, the slump test may not accurately represent the flowability of 3D-printed mortars under printing conditions. In this respect, the slump flow test is more suitable for characterizing the flowability of 3D mortars since it may account for the shear-thinning behavior [70]. This test set-up is similar to the slump test, but in this case, after the mold is removed, the flow table is dropped 25 times. The distance that the 3D mortar spreads was measured using a measuring tape (Figure 13). This measurement is known as the slump flow, which is the average diameter of the concrete spread. The values were correlated to the ones already imposed for the slump test, thus resulting in a slump flow range between 140 and 160 mm. These values were indicated in previous studies as suitable for 3D-printed mixtures [71].

## 6. Flexural Strength

Mechanical strength tests were performed on 3D-printed specimens made with the mixtures that proved to have satisfactory properties in the fresh state. The specimens either were extracted from blocks (in the case of the elements printed with the 20 mm nozzle) or directly printed into the desired shape (in the case of the elements manufactured using the 45 mm nozzle) based on the type of test to be conducted. The three-point bending tests were conducted on prism specimens at 24 h and 7, 14, and 28 days following the standard presented in [73]. For each specific time interval, five specimens manufactured with the M28 mix and printed with the 10 mm nozzle and five specimens manufactured with the M27 mix and printed with the 45 mm nozzle were tested (Figure 14 and Figure 15). The load rate was maintained at 0.1 kN/s during the test. The data were automatically recorded by a data logger and transferred to a computer. 

The variations in the flexural strengths at specific time intervals (24 h and 7, 14, and 28 days) are illustrated in Figure 16, Figure 17, Figure 18 and Figure 19.

## 7. Compressive Strength

The compressive tests were conducted on cubic specimens at 24 h and 7, 14, and 28 days following the standard presented in [74]. For each specific time interval, 10 specimens manufactured with the M28 mix and printed with the 10 mm nozzle and 5 specimens manufactured with the M27 mix and printed with the 45 mm nozzle were tested (Figure 20 and Figure 21). The load rate was maintained at 0.5 kN/s during the test. The data were automatically recorded by a data logger and transferred to a computer.

It should be noted that the surface quality might contribute to the mechanical performance, as mentioned in [72]. However, in this study, all of the surfaces of specimens were carefully polished and flattened before testing. Thus, the effect of the surface irregularities may be ignored. The variations in the compressive strengths at specific time intervals (24 h and 7, 14, and 28 days) are illustrated in Figure 22, Figure 23, Figure 24 and Figure 25.

## 8. Discussion

Based on the results presented in Table 2 (extrudability, buildability, and flowability), it was found that the M27 and M28 mixes possessed the most favorable properties for 3D printing. Consequently, mortar specimens were prepared and tested for their compressive and three-point bending strengths (Figure 26) using two nozzle dimensions (20 and 45 mm). 

The results indicate that the M27 mix printed with a 20mm nozzle had significantly higher values compared to the M28 mix printed with a 45mm nozzle. Moreover, the specimens made with the 45mm nozzle showed a higher susceptibility to eccentricities, leading to significantly lower compressive and flexural strength values compared to the specimens printed with a 20mm nozzle. Also, the compressive failure mechanism of specimens made with the M27 mix and printed with a 20mm nozzle was similar to the classic characteristic failure mode of cubic concrete specimens. Overall, the specimens made with the M28 mix and printed with the 20 mm nozzle have a flexural strength advantage of approximately 53%, 50%, 36%, and 32% over the specimens made with the M27 mix and printed with the 45 mm nozzle at 24 h, 7 days, 14 days, and 28 days, respectively (Figure 25). On the other hand, the differences recorded for the compressive strengths are even higher. Thus, the specimens made with the M28 mix and printed with the 20 mm nozzle have a compressive strength advantage of approximately 145%, 134%, 31%, and 60% over the specimens made with the M27 mix and printed with the 45 mm nozzle at 24 h, 7 days, 14 days, and 28 days, respectively (Figure 25). These findings suggest that the M28 mix with a 20mm nozzle is a better combination for 3D printing applications, as it provides remarkable properties in a fresh state and higher compressive and flexural strengths, which indicate a better structural stability compared to the specimens made with the M27 mix. 

Despite the success in developing the M28 mixture for 3D printing, the study identified some challenges that need to be further considered in future research. To address these challenges, the authors plan to print a prototype-scale structure using the M28 mix and the 20 mm nozzle. The prototype will be tested in dynamic/seismic mode, and reinforcing methods will be designed based on the authors’ previously developed studies related to fiber-reinforced polymer materials and textile-reinforced mortars (Figure 27) [75,76,77,78,79,80,81]. 

## 9. Conclusions and Future Research Directions

This paper presents the development of a fiber-reinforced mortar that can be 3D printed without extrusion. The study involved the preparation of 30 mixtures using three distinct preparation methods, as well as the preparation of 28 other mixtures after characterizing the preparation methods. Based on the experimental program described in this work, the following conclusions can be drawn:The M27 and M28 mixes have the most favorable fresh properties for 3D printing.Mixtures with lower stiffness exhibit improved flow characteristics, enabling consistent and uninterrupted material extrusion through the printing nozzle. Conversely, mixtures with excessive viscosity, such as those encountered in mixes M1-M16, pose significant challenges during the printing process. The high viscosity impedes the flow of the mortar material, making it prone to clogging within the printing nozzle. This clogging phenomenon disrupts the extrusion process, resulting in irregular material flow, the incomplete filling of printed layers, and a compromised structural integrity of the final printed components.In the case of mixes M27 and M28, the high buildability observed enabled the printing of multiple layers without any visible signs of instability or structural failure. This indicates that these mixtures possessed the necessary strength, viscosity, and bonding properties to sustain the progressive addition of layers and ensure the overall stability of the printed projects. The rest of the mixtures lacked the necessary strength, viscosity control, or bonding properties, leading to the progressive loss of stability or sudden plastic failure of the printed projects. As a result, the printed structures collapsed either gradually due to inadequate interlayer adhesion or suddenly due to an inability to bear the weight of the upper layers.Flowability within a range from 40 to 60 mm promotes the achievement of robust and structurally sound 3D-printed mortar structures. Mortar mixtures with lower flowability can result in inadequate compaction and compromised mechanical properties, while excessively high flowability may lead to reduced cohesion and interparticle interactions, thereby compromising the strength and durability of the printed components.The recommended slump flow range from 140 to 160 mm ensures favorable material flow behavior during 3D printing. Mixtures with slump flow values within this range exhibit appropriate viscosity and yield stress, facilitating consistent material flow through the printer nozzle. Slump flow values below 140 mm indicate a higher yield stress, making it challenging for the mortar to flow smoothly. On the other hand, slump flow values above 160 mm suggest excessive fluidity, increasing the risk of material spreading and the loss of structural stability during printing.The mechanical performance tests indicated that the 3D-printed specimens made with the M28 mix and printed through the 20 mm nozzle have considerably higher strengths than the ones made with the M27 mix and 45 mm nozzle. The improved strength of the mortar mixtures printed with narrow nozzles can be attributed to two primary mechanisms. Firstly, the reduced eccentricities minimize the formation of voids and weak spots within the printed layers, leading to improved structural integrity. Secondly, the enhanced exterior roundness achieved through the use of narrow nozzles ensures a more uniform distribution of forces during compression, resulting in higher strength values.Overall, this study presents a promising approach to 3D printing fiber-reinforced mortar without extrusion, and the results highlight the potential of using this technology for constructing complex structures with high strength and durability. The future research on this topic could lead to significant advances in the field of construction and infrastructure development. Also, the reduced facility requirements in this approach allow 3D printing to be made more available for civil engineering applications.

## Figures and Tables

**Figure 1 materials-16-04609-f001:**
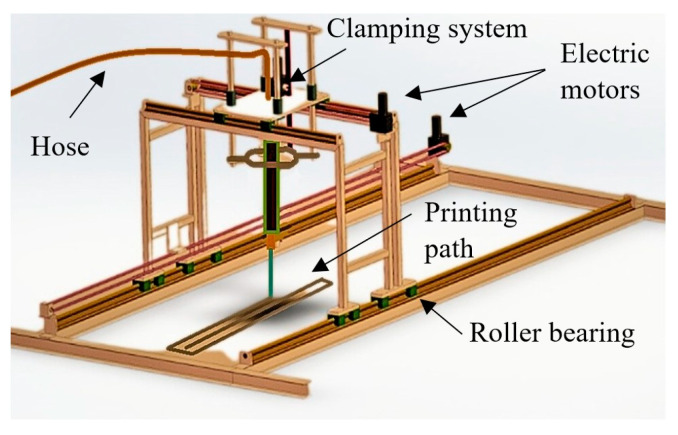
Configuration of the 3D gantry printer.

**Figure 2 materials-16-04609-f002:**
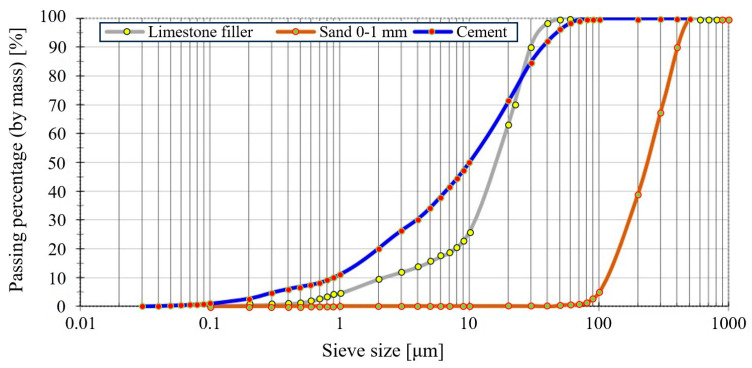
Particle size distribution.

**Figure 3 materials-16-04609-f003:**
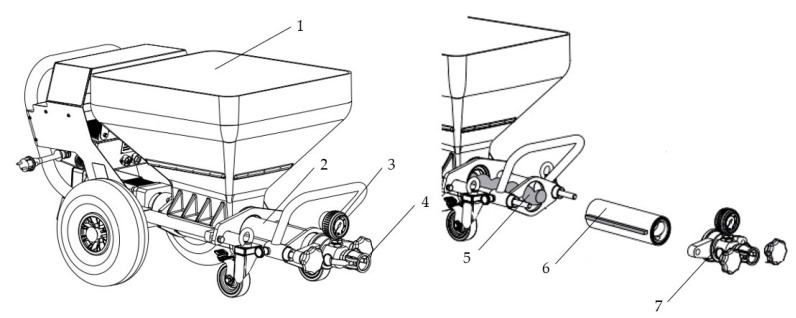
Mortar pump configuration: 1—container; 2—outlet unit with inside screw pump; 3—pressure gauge; 4—connection for hose; 5—rotor; 6—stator; 7—outlet unit.

**Figure 4 materials-16-04609-f004:**
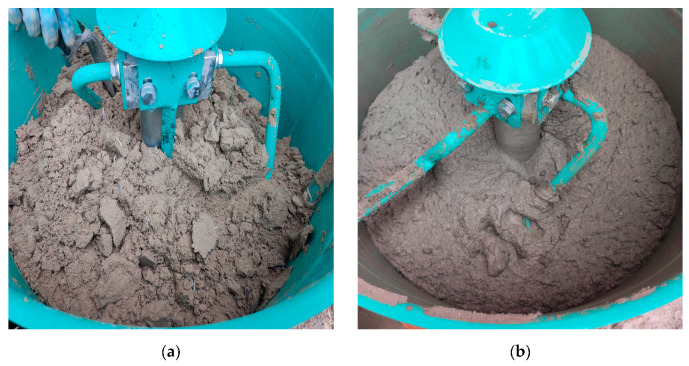
M28 mix (20 L charge) appearance and texture before (**a**) and after (**b**) the addition of the additives and final mixing.

**Figure 5 materials-16-04609-f005:**
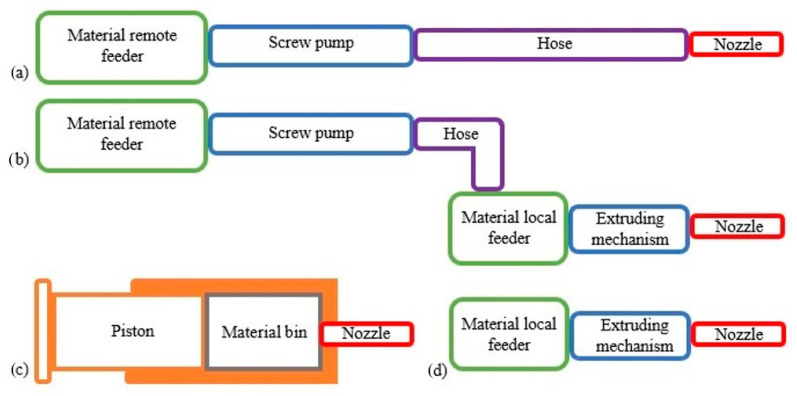
Material system and feeder configuration: (**a**) remote feeder, (**b**) remote and local feeder, (**c**) ram extruder, (**d**) local feeder.

**Figure 6 materials-16-04609-f006:**
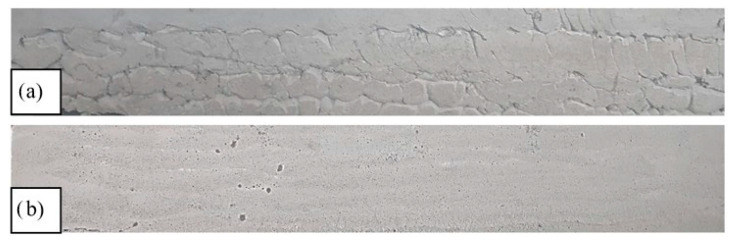
Lateral view of printed elements: (**a**) M4 mix—difficult extrusion (clogging issues), (**b**) M28 mix—smooth extrusion.

**Figure 7 materials-16-04609-f007:**
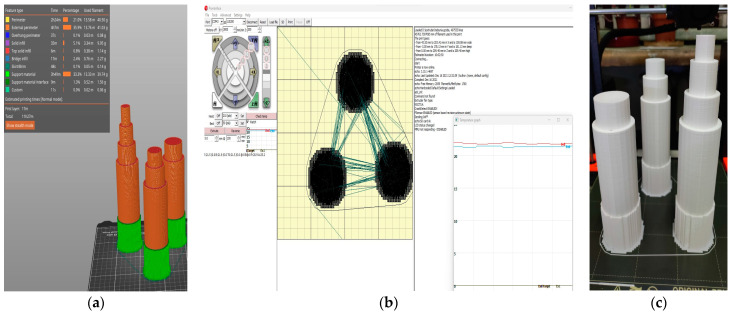
Nozzles: (**a**) 3D model, (**b**) printer monitor and control, (**c**) end products.

**Figure 8 materials-16-04609-f008:**
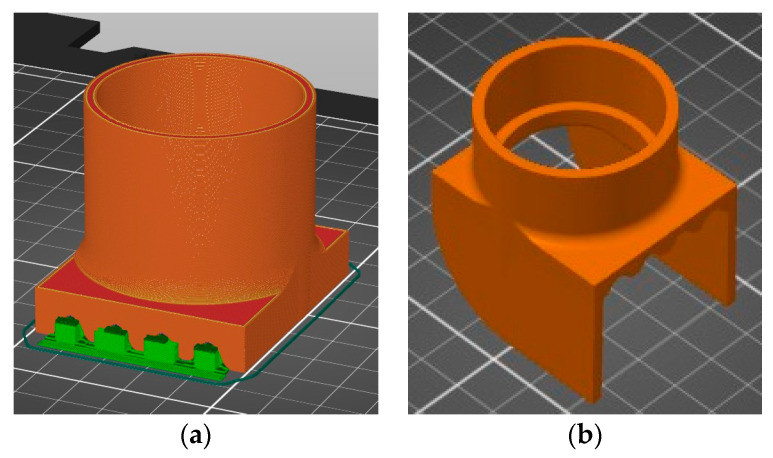
End accessories for the nozzles: (**a**) end cap that ensures a waved surface between the layers, (**b**) end cap that ensures smooth lateral surfaces for the printed elements.

**Figure 9 materials-16-04609-f009:**
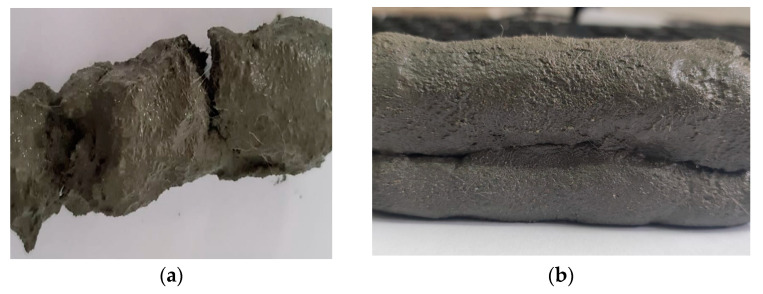
Three-dimensional layers: (**a**) collapsed layer by splitting (M10 mix), (**b**) layer with high buildability (M26 mix).

**Figure 10 materials-16-04609-f010:**
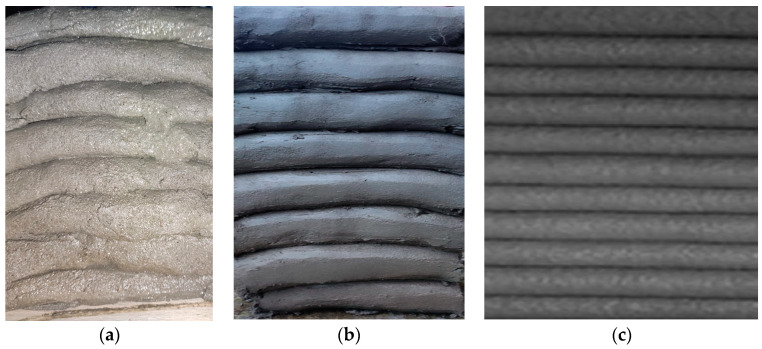
The buildability test: (**a**) M20 mix—photo taken just before collapse (large deformations occurred in the lower 3 layers), (**b**) M27 mix, 45 mm nozzle—high buildability (over 100 layers), (**c**) M28 mix, 20 mm nozzle—high buildability (over 100 layers).

**Figure 11 materials-16-04609-f011:**
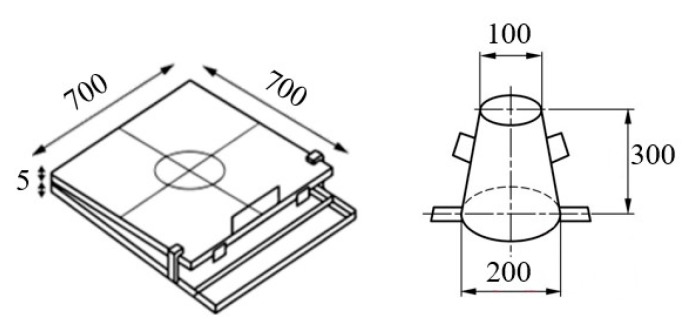
Flow table and conical mold. Dimensions in mm.

**Figure 12 materials-16-04609-f012:**
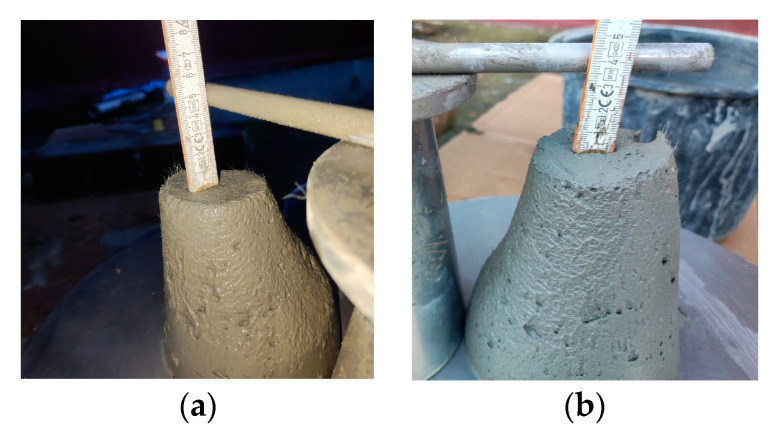
Slump test of printable mixtures (**a**) M27 and (**b**) M28.

**Figure 13 materials-16-04609-f013:**
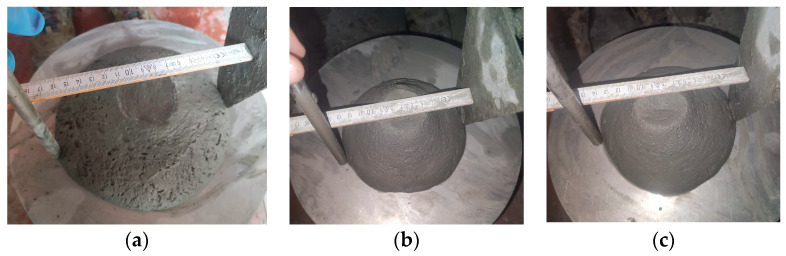
Slump flow test: (**a**) M17—very fluid mix, (**b**) M20—fluid mix, (**c**) M28—fluid and printable mix.

**Figure 14 materials-16-04609-f014:**
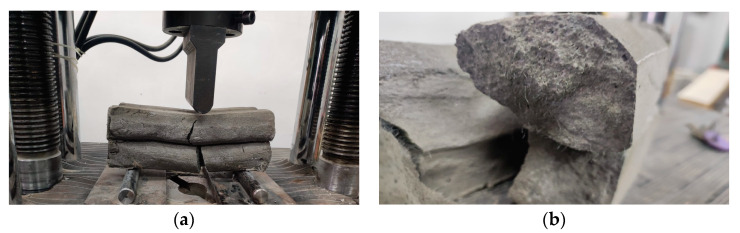
Specimen manufactured with M27 mix and printed with the 45 mm nozzle: (**a**) failure mode, (**b**) sectional view and orientation of fibers.

**Figure 15 materials-16-04609-f015:**
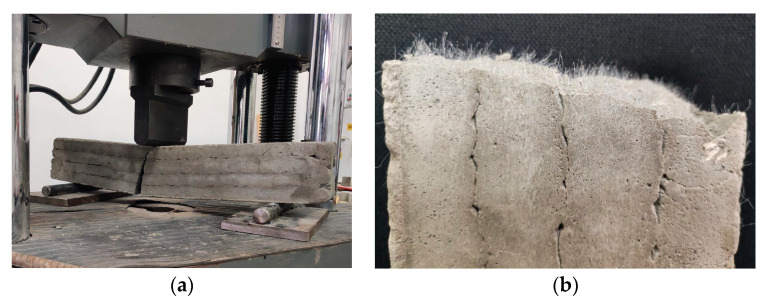
Specimen manufactured with M28 mix and printed with the 20 mm nozzle: (**a**) failure mode, (**b**) orientation of fibers.

**Figure 16 materials-16-04609-f016:**
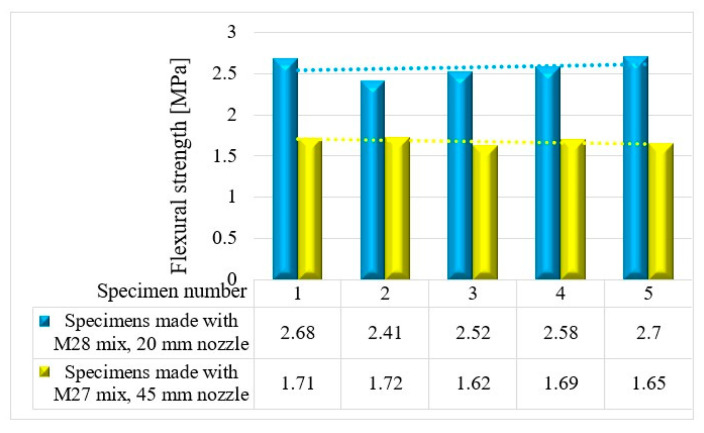
Flexural strengths—24 h.

**Figure 17 materials-16-04609-f017:**
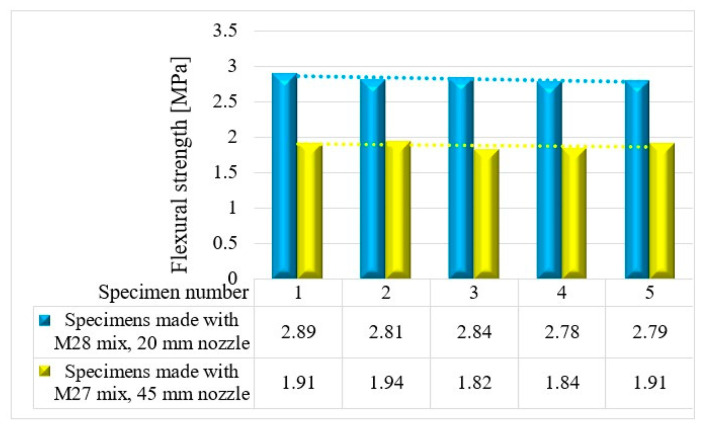
Flexural strengths—7 days.

**Figure 18 materials-16-04609-f018:**
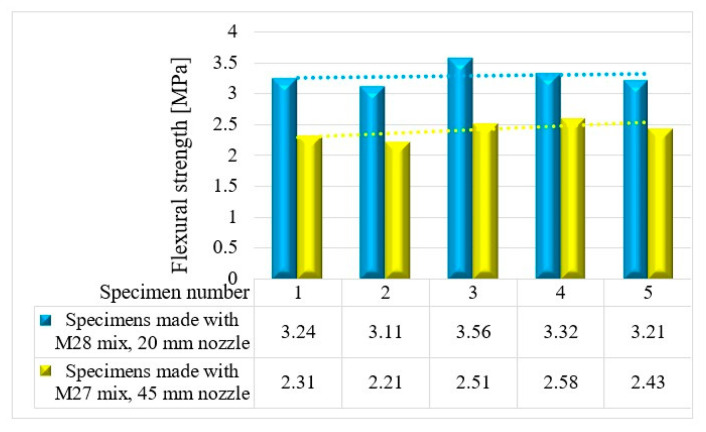
Flexural strengths—14 days.

**Figure 19 materials-16-04609-f019:**
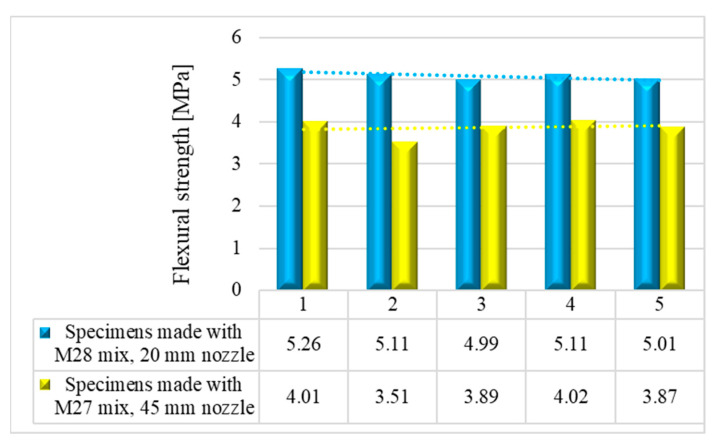
Flexural strengths—28 days.

**Figure 20 materials-16-04609-f020:**
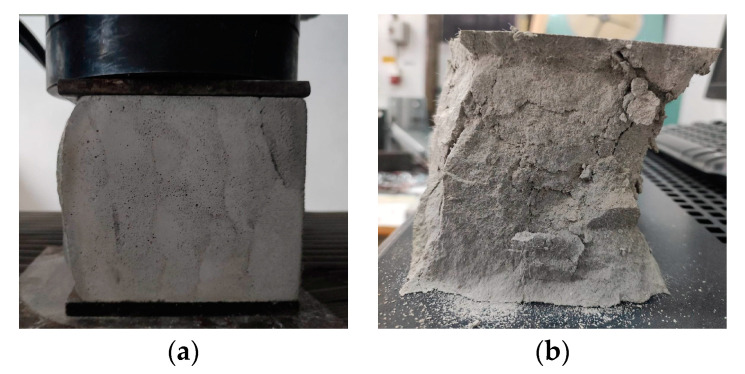
Specimen manufactured with M28 mix and printed with the 20 mm nozzle: (**a**) set-up, (**b**) failure mode.

**Figure 21 materials-16-04609-f021:**
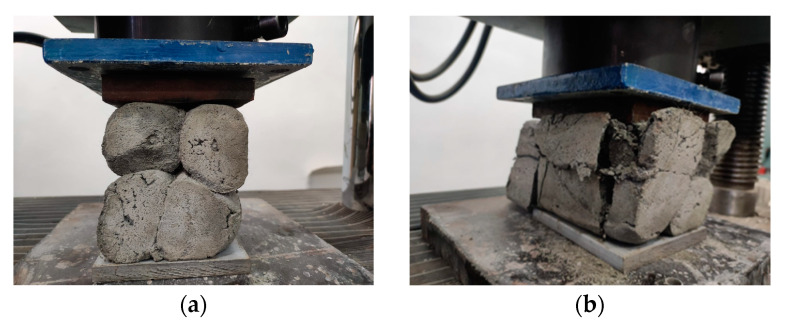
Specimen manufactured with M27 mix and printed with the 45 mm nozzle: (**a**) set-up, (**b**) failure mode.

**Figure 22 materials-16-04609-f022:**
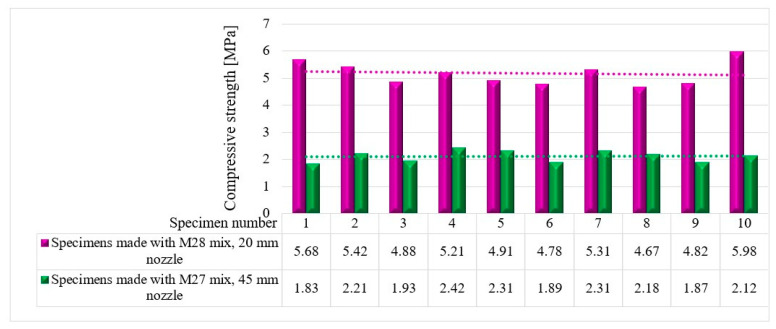
Compressive strengths—24 h.

**Figure 23 materials-16-04609-f023:**
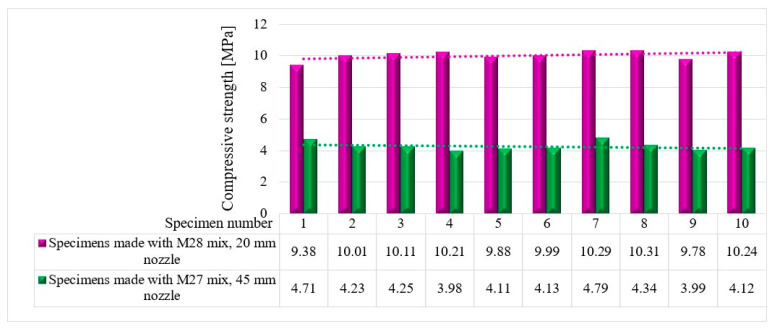
Compressive strengths—7 days.

**Figure 24 materials-16-04609-f024:**
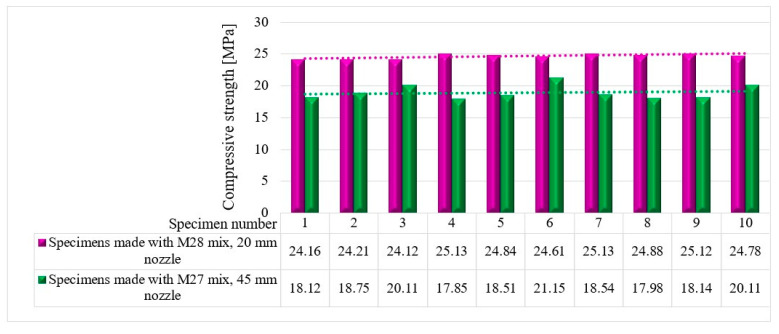
Compressive strengths—14 days.

**Figure 25 materials-16-04609-f025:**
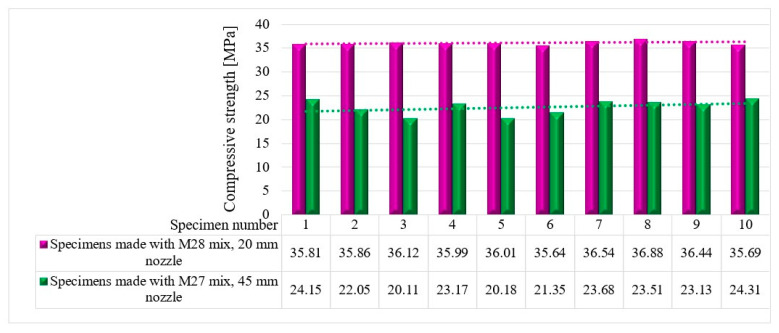
Compressive strengths—28 days.

**Figure 26 materials-16-04609-f026:**
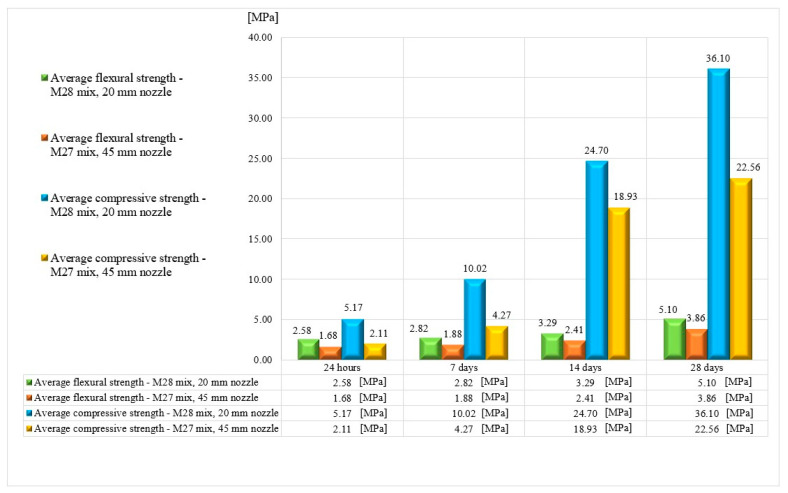
Average compressive and flexural strengths.

**Figure 27 materials-16-04609-f027:**
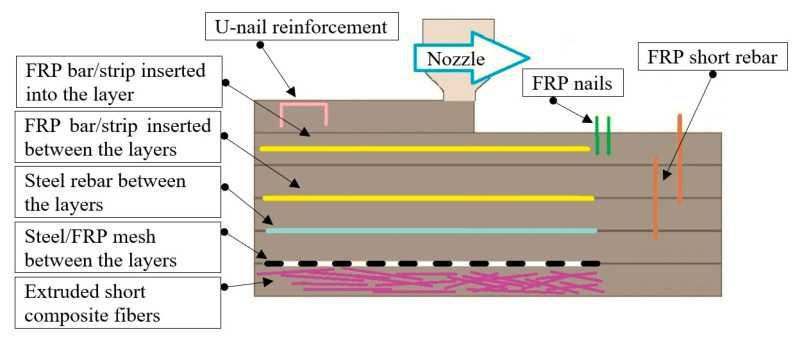
Proposed locations of the reinforcements in the 3D-printed mortar structure: in the layers, across the layers, and perpendicular to the layers.

**Table 1 materials-16-04609-t001:** Material mixing methods.

Method 1	Prepared in a Pan Mixer with Constant Speed
Step 1	Weigh all materials. This step is common for all of the methods; thus, it is not repeated in the next sections of this table.
Step 2	Mix the viscosity-modifying agent and the plasticizer in two separate glass containers with 1/3 water (calculated based on the agent’s mass) taken from the total water amount. This step is common for all of the methods; thus, it is not repeated in the next sections of this table.
Step 3	Mix the sand, the limestone filler, and the polypropylene fibers for 5 min.
Step 4	Add half of the water and mixing for 5 min.
Step 5	Let mixture settle for 5 min. This step is common for all of the methods; thus, it is not repeated in the next sections of this table.
Step 6	Add the cement and the rest of the water. Mix for 5 min.
Step 7	Add the viscosity-modifying agent and the plasticizer. Mix for 5 min.
Method 2	Prepared in a site concrete mixer with constant speed
Step 3	Mix the sand, the limestone filler, and the polypropylene fibers for 5 min. Manually disperse the fibers before adding them. During the mixing time, keep the drum constantly tilted between 30° and 50°.
Step 4	Add half of the water and mix for 5 min. During the mixing time, keep the drum constantly tilted between 30° and 50°.
Step 6	Add the cement and the rest of the water. Mix for 5 min.
Step 7	Add the viscosity-modifying agent and the plasticizer. Mix for 7 min.
Method 3	Prepared in a cylindrical tank using a handheld electrical mortar mixer with adjustable speed
Step 3	Mix the sand, the limestone filler, and the polypropylene fibers for 5 min. Gradually increase the speed up to 350 RPM. Use a cylindrical tank to avoid material trapping at corners.
Step 4	Add half of the water and mix for 5 min at 500 RPM.
Step 6	Add the cement and the rest of the water. Mix for 5 min at 700 RPM.
Step 7	Add the viscosity-modifying agent and the plasticizer. Mix for 7 min at 700 RPM.

**Table 2 materials-16-04609-t002:** Mix formulations—quantities per cubic meter.

Mix	Sand (kg)	Cement (kg)	Limestone Filler (kg)	Fibers (kg)	Superplasticizer (%)	Viscosity Modifying Agent (%)	Water (L)	W/C
**M1**	1358	580	200	7	1.2	0.2	200	0.345
**Extrudability**		The M1 mix could not be extruded. Blockage occurred in the pump feeder shaft.						
**M2**	1358	580	200	5	1.2	0.2	200	0.345
**Extrudability**		The M2 mix could not be extruded. Blockage occurred in the pump feeder shaft.						
**M3**	1358	580	200	3	1.2	0.2	200	0.345
**Extrudability**		The M3 mix could not be extruded. Blockage occurred either in the pump feeder shaft or in the outlet unit.						
**M4**	1358	580	200	1	1.2	0.2	200	0.345
**Extrudability**		The M4 mix could be extruded but blockages still occurred. Thus, the water quantity was gradually increased starting from a step of 5 l/m^3^. The final quantity of water (265 l) corresponded to the M17 mix.						
**M17**	1358	580	200	1	1.2	0.2	265	0.457
**Extrudability**		The M17 mix could be extruded without blockages.						
**Buildability**		The M17 mixture could not be printed as large deformations occurred in the bottom layers. The mix was too fluid; the percentage of viscosity-modifying agent was increased by 0.1%.						
**M18**	1358	580	200	1	1.2	0.3	265	0.457
**Extrudability**		The M18 mix could be extruded without blockages.						
**Buildability**		The M18 mixture could not be printed as large deformations occurred in the bottom layers. The mix was too fluid; the percentage of viscosity-modifying agent was increased by 0.1%.						
**M19**	1358	580	200	1	1.2	0.4	265	0.457
**Extrudability**		The M19 mix could be extruded without blockages.						
**Buildability**		The M19 mixture could not be printed as large deformations still occurred in the bottom layers. The mix was too fluid; the percentage of viscosity-modifying agent was increased by 0.2%.						
**Slump flow**		150 mm						
**M20**	1358	580	200	1	1.2	0.6	265	0.457
**Extrudability**		The M20 mix could be extruded without blockages.						
**Buildability**		The M20 mixture could not be printed as large deformations still occurred in the bottom layers. The mix was too fluid; the percentage of viscosity-modifying agent was increased by 0.2%.						
**Slump flow**		140 mm						
**M21**	1358	580	200	1	1.2	0.8	265	0.457
**Extrudability**		The M21 mix could be extruded without blockages.						
**Buildability**		The M21 mixture could be printed.						
**Slump flow**		110 mm						
**Printability**		The open time of the M21 mix was too high. Thus, the quantity of water was gradually reduced by 5 L/m^3^. The final quantity of water (245 L) corresponded to the M25 mix. Also, the percentage of plasticizer was increased by 0.2%.						
**M25**	1358	580	200	1	1.4	0.8	245	0.422
**Extrudability**		The M25 mix could not be extruded. Blockage occurred either in the pump feeder shaft or in the outlet unit. The quantity of water was gradually increased by 5 L/m^3^. The final quantity of water (255 L) corresponded to the M27 mix. The plasticizer was reduced by 0.2%.						
**Slump flow**		135 mm						
**M27**	1358	580	200	1	1.2	0.4	255	0.440
**Extrudability**		The M27 mix could be extruded only through the 45 mm nozzle.						
**Buildability**		The M27 mixture could be printed.						
**Printability**		The open time of the M27 mix was around 35 min.						
**M28**	1358	580	200	1	1.1	0.4	265	0.457
**Extrudability**		The M28 mix could be extruded through the 18, 20, 25, and 45 mm nozzles.						
**Buildability**		It was found that the M28 mix can build more than 100 layers without showing any type of failure.						
**Printability**		The open time of the M28 mix was around 40 min. The optimum printing speed was limited to 100 mm/s to print layers with the same width as the nozzle’s smallest inlet (18 mm).						
**Slum flow**		160 mm						
**Slump**		40 mm						
**Density**		2249 kg/m^3^						

## Data Availability

The data underlying this article will be shared upon reasonable request from the corresponding author.

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
