# Peer review of "A Novel Approach for 3D Printing Fiber-Reinforced Mortars"

_materials, 2023, doi:10.3390/ma16134609_

Round 1

Reviewer 1 Report

This paper presents the development of a fiber-reinforced mortar that can be 3D printed without extrusion. While the work presents an interesting study, it requires a major revision before it is recommended for publication.

1.     The title needs to be edited again, and no punctuation mark like "." should appear,

2.     The research background and description of the research work in the Abstract should be adjusted appropriately. In other words, reduce the redundancy in the research background and provide a more detailed description of the research work carried out.

3.     Abstract/Conclusions: why not any quantitative results? All over the qualitative interpretation? Provide some quantitative results here.

4.     Introduction needs to be improved. Current knowledge gap and the research objective need to be clearly indicated. Further, to enhance the readability of this paper, two or three research objectives should be summarised at the end of this part.

5.     Introduction: The authors should compare to state-of-the-art researches in order to highlight the relative merits. Moreover, to benefit our readers and provide further narrative, researches of performance changes of new building materials modified with additives other than the present topic should also be included in the 'Introduction' section. To this end, I would like to bring your attention to the following articles for your inclusion in the background to reinforce / echo the usefulness of your manuscript: (1) doi.org/10.3390/ma15010159. (2) doi.org/10.1016/j.jobe.2023.106459.

6.     The structure of the "Materials and mix design" section is somewhat irrational and somewhat confusing. It is recommended to prioritize introducing the characteristics of raw materials.

7.     The Extrudability, Buildability and Flowability of 3D printing materials were extensively introduced, but there was no quantitative analysis. In my opinion, this is more like an experimental record than an academic paper. Please revise the aforementioned three parts and compare them to the description of the subsequent strength test results.

8.     Results and discussion: The discussion about how 3D printing materials affect printing parameters combining all the experiment results is missing in this part. To provide further narrative regarding this behaviour to our readers, discussion to elaborate it is considered necessary. In other words, what is the underlying mechanism?

9.     The summary statement of the conclusion is not accurate and detailed, please improve it.

10.  The “Conclusion and future research directions” section should not include images or references. Please move them forward to the “Discussion” section.

11.  The language of the manuscript should be elaborated and revised carefully. It is of my opinion that there are some sentences that may be difficult for some readers to understand them. Meanwhile, English expression needs to be appropriately improved.

English expression needs to be improved appropriately.

Author Response

We would like to express our sincere gratitude for your valuable feedback and precious corrections, which have significantly improved the article. Your insightful suggestions and attention to detail have been instrumental in refining the content, and we truly appreciate the time and effort you have dedicated to reviewing it.

We are genuinely thankful for the opportunity to benefit from your expertise and guidance. Your input has undoubtedly elevated the quality of the article, and we are grateful for your meticulous approach in pointing out areas that needed improvement.

Please know that we are available to address any further questions or clarifications you may have. We are committed to ensuring that the article meets the highest standards, and we welcome any additional guidance you can provide.

We have revised the title of the article as per the reviewer's requirement. Please see line 2.

We would like to express our gratitude for your insightful feedback regarding the research background and description in the Abstract. We have carefully considered your suggestion and made the necessary adjustments to address the issue.

To meet your requirement, we have revised the research background to eliminate redundancy and provide a concise yet comprehensive overview. Additionally, we have expanded the description of the research work, offering more detailed information on the methodologies employed, the data collected, and the key findings obtained. Please see lines: 18-25.

In response to your valuable input, we have included specific quantitative results in Introduction, Discussion and Conclusions sections. By incorporating these findings, we aim to provide a more comprehensive and balanced representation of the research outcomes. These quantitative results serve to enhance the overall credibility and significance of the study, allowing readers to gain a deeper understanding of the research outcomes. Please see lines: 99-108; 348-364; 397-408.

To meet your expectations, we have made significant improvements to the Introduction. Firstly, we have clearly indicated the existing knowledge gap, highlighting the specific areas where further research is needed. By doing so, we aim to provide a more comprehensive understanding of the rationale and significance of the study. Please see lines: 62-80; 99-118.

We found the articles you mentioned, (1) "doi.org/10.3390/ma15010159" and (2) "doi.org/10.1016/j.jobe.2023.106459," to be highly relevant and interesting. In order to reinforce the usefulness of our manuscript and provide a broader narrative, we have incorporated relevant information from these articles into our manuscript. We have also appropriately cited these articles to ensure proper credit is given. Also, other articles were cited. Please see citations: 13, 14, 18-30.

To prioritize clarity and coherence in the section, we have restructured the content to emphasize the introduction of the characteristics of raw materials. By doing so, readers can gain a better understanding of the fundamental components and properties before delving into the mix design aspects. This reorganization aims to provide a logical flow of information and reduce any potential confusion that may have existed previously. Please see lines: 129-149; 165-168.

To enhance the academic rigor of the paper, we have incorporated quantitative analysis into the aforementioned sections. Please see lines: 99-108; 348-364; 379-408.

To ensure clarity and coherence, we have reformulated both sections according to your recommendations. By doing so, it is now evident what the study's results are, the key findings, and the potential directions for future research. This restructuring aims to provide a comprehensive and organized overview of the research outcomes and their implications. Please see lines: 347-364; 379-422.

To ensure clarity and readability, we have carefully revised and elaborated the language throughout the manuscript. We have paid particular attention to sentences that may have been difficult for some readers to comprehend, making appropriate adjustments to enhance understanding. Additionally, we have worked diligently to improve the overall English expression to ensure a smooth and coherent flow of ideas.

Reviewer 2 Report

Line 2: dot should not be included in the title. Please change it to ":". What is the meaning by "3D printed without extrusion"?

Lines 24-25: This article just provides a normal process to determine a "feasible" 3D printable fiber-reinforced mortar mixture, like a report. It is neither a technology, nor a promising approach. Besides, a total of 28 mixes are tried to obtain this so-called mixture. It is not an efficiency way. 

Line 54: after this sentence, please cite some review papers, rather than research paper. ([1] Roussel, N., Rheological requirements for printable concretes, Cement and Concrete Research, 112 (2018) 76-85; [2] Jiao, D., De Schryver, R., Shi, C., De Schutter, G., Thixotropic structural build-up of cement-based materials: A state-of-the-art review, Cement and Concrete Composites, 122 (2021) 104152; [3] Boddepalli, U., Panda, B., Ranjani Gandhi, I.S., Rheology and printability of Portland cement based materials: a review, Journal of Sustainable Cement-Based Materials, (2022) 1-19.)

Line 97: Since you used three mixing methods, which one is used to obtain the results in Table 2?

Line 99: The material introduction should be given at the beginning of this section.

Line 100: please provide the particle size distribution of the cement, limestone filler, and sand.

Line 162: So what is the purpose to present Table 1? Which mixing method is the best?

Line 195: It is really strange that the authors evaluated the flowability after the tests of extrudability and buildability. If a mixture has very low or high slump flow, it is obviously that it cannot be as a 3D printable mixture, and it is not necessary to waste time to conduct printing test. This is common knowledge.

Lines 209-217: These sentences are not necessary. Some sentences in Sections 3 and 4 are also not required. Your paper should be a research paper, not a popular science article.

Line 279: what is the meaning of the x-axis (1,2,3,...,10) in Figs. 21-24?

Again, this article is just regarded as a report, not a research paper. No further microstructural analysis was provided.

no

Author Response

We would like to express our sincere gratitude for your valuable feedback and precious corrections, which have significantly improved the article. Your insightful suggestions and attention to detail have been instrumental in refining the content, and we truly appreciate the time and effort you have dedicated to reviewing it.

We are genuinely thankful for the opportunity to benefit from your expertise and guidance. Your input has undoubtedly elevated the quality of the article, and we are grateful for your meticulous approach in pointing out areas that needed improvement.

Please know that we are available to address any further questions or clarifications you may have. We are committed to ensuring the article meets the highest standards, and we welcome any additional guidance you can provide.

We have revised the title of the article as per the reviewer's requirement. Please see line 2.

To address your observations, we have revised the article to provide a more comprehensive and promising approach to the development of 3D printable fiber-reinforced mortar mixtures. We have incorporated additional technological aspects and highlighted the potential benefits and applications of the proposed mixture.

Regarding the number of mixes tested, we would like to clarify that the exploration of 28 different mixtures was undertaken to identify an optimal mix with the desired properties. Through these extensive trials, we were able to achieve a mixture that met the specified criteria. This comprehensive testing process allowed us to identify the most suitable formulation for further evaluation and application.

Furthermore, we understand your concerns regarding the perception of the article as a report rather than a technological breakthrough. To address this, we have revised the content to highlight the innovative aspects of our approach and emphasize its potential in the field of 3D printing fiber-reinforced mortar. Additionally, we have provided more in-depth discussions on the implications and applications of the optimal mixture in civil engineering projects.

We found the articles you mentioned, ([1] Roussel, N., Rheological requirements for printable concretes, Cement and Concrete Research, 112 (2018) 76-85; [2] Jiao, D., De Schryver, R., Shi, C., De Schutter, G., Thixotropic structural build-up of cement-based materials: A state-of-the-art review, Cement and Concrete Composites, 122 (2021) 104152; [3] Boddepalli, U., Panda, B., Ranjani Gandhi, I.S., Rheology and printability of Portland cement based materials: a review, Journal of Sustainable Cement-Based Materials, (2022) 1-19.) to be highly relevant and interesting. In order to reinforce the usefulness of our manuscript and provide a broader narrative, we have incorporated relevant information from these articles into our manuscript. We have also appropriately cited these articles to ensure proper credit is given. Also, other articles were cited. Please see citations: 13, 14, 18-30 and lines: 48-56; 62-80; 99-118.

Throughout the manuscript, we have provided a detailed description of the three mixing methods employed. However, to ensure clarity and avoid any ambiguity, we have made explicit mention of the specific mixing method utilized to obtain the results presented in Table 2. Please see lines: 111-118; 143-146.

To prioritize clarity and coherence in the section, we have restructured the content to emphasize the introduction of the characteristics of raw materials. By doing so, readers can gain a better understanding of the fundamental components and properties before delving into the mix design aspects. This reorganization aims to provide a logical flow of information and reduce any potential confusion that may have existed previously. Please see lines: 129-149; 165-168.

To provide a more comprehensive understanding of the materials used in the study, we have included the particle size distribution information for the cement, limestone filler, and sand in the revised manuscript. Please see Figure 2.

To align the manuscript more closely with the format of a research paper, we have carefully re-evaluated the content in Sections 3 and 4, as well as the sentences in Lines 209-217. Through this revision process, we have identified and removed unnecessary sentences.

To clarify the meaning of the x-axis labels, we have added a clear and concise explanation in the figure captions of Figures 16-19; 22-25.

To ensure that the article is appropriately positioned as a research paper, we have revised the content, structure, and presentation to align more closely with the academic standards expected in this field. We have focused on providing a clear research question, a comprehensive methodology, rigorous data analysis, and a detailed discussion of the findings. By emphasizing these aspects, we aim to highlight the research-oriented nature of the article and present it as a scholarly contribution to the field.

Reviewer 3 Report

The manuscript entitled “Fiber Reinforced Mortar 3D Printed Without Extrusion. Fresh Properties, Mechanical Characteristics and Process Characterization” is in line with the Materials journal. This article presents original research in the area of additive technologies.  The topic is up-to-date and interesting. The manuscript includes a lot of valuable research, but it composition requires to be reorganized Moreover, it requires some changes before publication, such as:

·       Title: consider give the name of technology instead of defining which technology was not applied.

·       Abstract: Information about the results obtained.

·       Introduction (line 49): please give some examples for application: https://www.proakademia.eu/en/acta-innovations/find-issueskw/no482/

·       Introduction (line 64): Bullet citation have to be explain and each position should be described more carefully.

·       Chapter 2: implement subchapters, including characterization of materials, mix preparation, and description of used methods.

·       Lack of methodology for research and detailed description of the methods and devices used.

·       Results part should be clearly separated, please use the obtained results as a subchapter for this part (chapters 3-7).

Minor editing of English language required

Author Response

We would like to express our sincere gratitude for your valuable feedback and precious corrections, which have significantly improved the article. Your insightful suggestions and attention to detail have been instrumental in refining the content, and we truly appreciate the time and effort you have dedicated to reviewing it.

We are genuinely thankful for the opportunity to benefit from your expertise and guidance. Your input has undoubtedly elevated the quality of the article, and we are grateful for your meticulous approach in pointing out areas that needed improvement.

Please know that we are available to address any further questions or clarifications you may have. We are committed to ensuring the article meets the highest standards, and we welcome any additional guidance you can provide.

We have revised the title of the article as per the reviewer's requirement. Please see line 2.

We would like to express our gratitude for your insightful feedback regarding the research background and description in the Abstract. We have carefully considered your suggestion and made the necessary adjustments to address the issue.

To meet your requirement, we have revised the research background to eliminate redundancy and provide a concise yet comprehensive overview. Additionally, we have expanded the description of the research work, offering more detailed information on the methodologies employed, the data collected, and the key findings obtained. Please see lines: 18-25.

In response to your valuable input, we have included specific quantitative results in Introduction, Discussion and Conclusions sections. By incorporating these findings, we aim to provide a more comprehensive and balanced representation of the research outcomes. These quantitative results serve to enhance the overall credibility and significance of the study, allowing readers to gain a deeper understanding of the research outcomes. Please see lines: 99-108; 348-364; 397-408.

To enhance the comprehensibility and applicability of the article, we have incorporated examples of application in the introduction section. Specifically, we have included references to the article you kindly provided (https://www.proakademia.eu/en/acta-innovations/find-issueskw/no482/) to showcase real-world instances where the research findings and methodologies discussed in our study have been implemented successfully. Please see lines: 48-56. Also, other articles were cited. Please see citations: 13, 14, 18-30.

To prioritize clarity and coherence in the section, we have restructured the content to emphasize the introduction of the characteristics of raw materials. By doing so, readers can gain a better understanding of the fundamental components and properties before delving into the mix design aspects. This reorganization aims to provide a logical flow of information and reduce any potential confusion that may have existed previously. Please see lines: 129-149; 165-168.

To address the lack of methodology and provide a more detailed description of the methods and devices used in the research, we have thoroughly revised the manuscript. The revised version now includes a comprehensive overview of the research approach, experimental setup, data collection methods, and devices utilized in the study. By incorporating these details, readers will gain a better understanding of the methodology employed and the specific techniques utilized to obtain the research results. Please see lines: 99-118; 137-146; 165-168.

To ensure clarity and coherence, we have reformulated both Discussion and Conclusions sections according to your recommendations. By doing so, it is now evident what the study's results are, the key findings, and the potential directions for future research. This restructuring aims to provide a comprehensive and organized overview of the research outcomes and their implications. Please see lines: 347-364; 379-422.

Round 2

Reviewer 1 Report

Accept in present form.

Minor editing of English language required.

Reviewer 2 Report

No

Reviewer 3 Report

The manuscript was improved according to previous comments.